# Phosphorylation but Not Oligomerization Drives the Accumulation of Tau with Nucleoporin Nup98

**DOI:** 10.3390/ijms23073495

**Published:** 2022-03-23

**Authors:** Lisa Diez, Larisa E. Kapinos, Janine Hochmair, Sabrina Huebschmann, Alvaro Dominguez-Baquero, Amelie Vogt, Marija Rankovic, Markus Zweckstetter, Roderick Y. H. Lim, Susanne Wegmann

**Affiliations:** 1German Center for Neurodegenerative Diseases (DZNE), Charitéplatz 1, 10117 Berlin, Germany; lisa.diez@dzne.de (L.D.); janine.hochmair@dzne.de (J.H.); sabrina.huebschmann@dzne.de (S.H.); alvaro.dominguez-baquero@dzne.de (A.D.-B.); am.vogt97@gmail.com (A.V.); 2Biozentrum and the Swiss Nanoscience Institute, University of Basel, 4056 Basel, Switzerland; larisa.kapinos@unibas.ch (L.E.K.); roderick.lim@unibas.ch (R.Y.H.L.); 3Max Planck Institute for Multidisciplinary Sciences, Am Fassberg 11, 37077 Goettingen, Germany; mara@nmr.mpibpc.mpg.de (M.R.); mazw@mpinat.mpg.de (M.Z.); 4German Center for Neurodegenerative Diseases (DZNE), Von-Siebold-Str. 3a, 37075 Goettingen, Germany

**Keywords:** MAPT, nuclear pore complex, FG-Nups, posttranslational modifications

## Abstract

Tau is a neuronal protein that stabilizes axonal microtubules (MTs) in the central nervous system. In Alzheimer’s disease (AD) and other tauopathies, phosphorylated Tau accumulates in intracellular aggregates, a pathological hallmark of these diseases. However, the chronological order of pathological changes in Tau prior to its cytosolic aggregation remains unresolved. These include its phosphorylation and detachment from MTs, mislocalization into the somatodendritic compartment, and oligomerization in the cytosol. Recently, we showed that Tau can interact with phenylalanine-glycine (FG)-rich nucleoporins (Nups), including Nup98, that form a diffusion barrier inside nuclear pore complexes (NPCs), leading to defects in nucleocytoplasmic transport. Here, we used surface plasmon resonance (SPR) and bio-layer interferometry (BLI) to investigate the molecular details of Tau:Nup98 interactions and determined how Tau phosphorylation and oligomerization impact the interactions. Importantly, phosphorylation, but not acetylation, strongly facilitates the accumulation of Tau with Nup98. Oligomerization, however, seems to inhibit Tau:Nup98 interactions, suggesting that Tau-FG Nup interactions occur prior to oligomerization. Overall, these results provide fundamental insights into the molecular mechanisms of Tau-FG Nup interactions within NPCs, which might explain how stress-and disease-associated posttranslational modifications (PTMs) may lead to Tau-induced nucleocytoplasmic transport (NCT) failure. Intervention strategies that could rescue Tau-induced NCT failure in AD and tauopathies will be further discussed.

## 1. Introduction

The axonal microtubule-associated protein Tau (MAPT) is predominantly found in neurons of the central nervous system. The best-known function of Tau is the binding and stabilization of microtubules (MT), which is regulated by Tau phosphorylation [1]. In addition, Tau is closely linked to multiple neurodegenerative diseases (NDDs) including Alzheimer’s disease (AD), frontotemporal dementia (FTD), Pick’s disease, and Progressive supranuclear palsy. In these diseases, phosphorylated Tau accumulates and aggregates in the cell body of neurons, which is associated with neuronal loss and progressive cognitive impairment [2,3]. However, Tau-induced neurotoxicity appears to occur prior to aggregation and to be mediated by soluble forms of Tau [4,5].

Tau phosphorylation and redistribution into the somatodendritic compartment and into the nucleus occur upon neuronal stress exposure [6,7] and are seen as early changes related to Tau neurotoxicity in AD [8,9,10]. Upon missorting, Tau can engage in a number of ‘abnormal’ interactions and functions that may drive neurotoxicity prior to Tau aggregation [11]. Among such disease- and stress-associated interactions of Tau is its accumulation at the outer nuclear envelope (NE) [12,13]. In addition, deformations of the NE (lamin folds and invaginations), can be observed in AD brains [14], Tau transgenic flies [15], and in neurons in vitro [16]. Of note, NE deformations and dysfunctions also occur in normal aging [17].

A consequence of defective NE and Tau accumulation at the NE is the impairment of the nucleocytoplasmic transport (NCT), the molecular transport into and out of the nucleus. Notably, NCT is an essential process for most cellular functions [18,19]. For example, the nuclear import of transcription factors is needed to regulate all transcriptional events, and the NCT of mRNA and ribosomal proteins and pre-complexes is essential for translation. The NE ensures the separation of the nucleoplasm and cytoplasm, and the exchange of molecules between the two compartments is regulated by nuclear pore complexes (NPC) embedded in the NE. NPCs are megadalton complexes made of >30 proteins, called nucleoporins (Nups), and regulate the exchange of RNAs and proteins. Small molecules can freely diffuse through NPCs, whereas nuclear transport receptors (NTRs) are required to shuttle molecules larger than ~30 kDa in mass or ~5 nm in diameter through the pore. Structural ‘scaffold’ Nups build the NPC ring upon which cytoplasmic ring/filament Nups are attached. Within the central channel, several nucleoporins harbor phenylalanine–glycine (FG) repeats, the so-called FG-Nups, which mediate NPC cargo translocation by binding NTRs. Otherwise, the intrinsically disordered N-termini of FG-Nups, such as Nup98, exhibit dynamic fluctuations that establish the NPC permeability barrier [20]. Importantly, somatodendritic Tau was shown to block both mRNA [21] and protein [12] NCT in neurons, and it was suggested that the direct interaction of Tau with Nups can induce NCT impairment in AD [12]. Notably, Tau was shown to interact with certain central channel FG-Nups, i.e., Nup98 and Nup62, in different ways [12]: The flexible N-terminal part of certain central channel FG-Nups, i.e., Nup98 and Nup62, interacts with soluble Tau in vitro, co-immunoprecipitated with Tau from human AD brain lysates, and accumulates in Tau neurofibrillary tangles in AD brains. The negatively charged C-terminal half of Nup98, which is usually buried within the NPC scaffold, co-aggregates with recombinant Tau in vitro. Although these findings suggest that Tau interactions with the NPCs may induce pathological NCT failure early in disease (Figure 1A), molecular details of Tau interactions with FG-Nups remain unknown.

Tau itself is a highly soluble, intrinsically disordered protein with an inhomogeneous charge distribution. Human full-length Tau (441 aa, 2N4R isoform) comprises four major domains (Figure 1B): The acidic N-terminal half with two N-terminal inserts (N1, N2) of unknown function that can be alternatively spliced to create shorter (0N and 1N) isoforms; two proline-rich domains (P1 and P2) that carry many of Tau’s phosphorylation sites; a repeat domain with four ∼30 aa-long pseudo-repeats (R1-R4), and a short (40 aa) C-terminal end. The positively charged (at pH 7–8) repeat domain together with the adjacent regions (P2 and R’) mediate the binding of Tau to MTs [22,23,24,25], whereas the N-terminal half projects away from the MT surface (projection domain). The repeat domain, TauRD, is also responsible for Tau’s aggregation and builds the core of Tau fibrillar aggregates [26,27,28,29,30,31].

In this study, we aimed to decipher the molecular mechanism and the kinetics of Tau’s binding to Nup98. Using surface plasmon resonance (SPR) and biolayer interference (BLI), we characterized the binding of different recombinant human Tau variants to the intrinsically disordered Nup98 N-terminal half, Nup98-FG. Our data show that the Tau repeat domain mediates the binding of Tau to Nup98-FG and that phosphorylation but not oligomerization changes the binding kinetics of Tau and facilitates its accumulation on the Nup98-FG layer. These findings are reminiscent of observations in human AD brains, in which we detected accumulation of phospho-Tau at the NE in neurons without neurofibrillary tangles (NFTs).

## 2. Results

### 2.1. Tau Accumulates at Neuronal NEs in AD Brains

One of the early changes in Tau observed in AD is the mislocalization of phospho-Tau (P-Tau) to the somatodendritic compartment (Figure 1A), which occurs prior to Tau aggregation and the formation of neurofibrillary tangles (NFTs) [6]. In AD brains with substantial Tau pathology (Braak 5/6), nuclei isolated from frontal cortex grey matter were enriched in Tau phosphorylated at Ser396 and Ser404 (PHF-1 epitope; Figure 1C). Immunofluorescence microscopy showed that P-Tau (pT231 and pS396/pS404) was present in the soma and accumulated at the NE around nuclei of neurons without NFTs (Figure 1D). These data indicate that Tau enriches neuronal nuclei of AD brains.

### 2.2. Human Tau Binds to Nup98-FG

Prior studies suggested that a direct interaction of Tau with FG-Nups, such as Nup98, leads to NCT impairment [12]. To reveal the molecular determinants of this interaction, we tested the binding of Tau to Nup98-FG in vitro using SPR, a method that allows us to measure binding affinities and kinetic parameters of soluble Tau molecules (analyte, A) binding to a tethered layer of Nup98-FG (ligand, L; [32]) (Appendix A). Nup98-FG is comprised of the unstructured N-terminal half of Nup98 (aa 1-498) that forms, together with other FG-Nups, the permeability barrier in the central pore of NPCs. Immobilization of Nup98-FG on gold-coated SPR chips was achieved through cysteine residues inserted at the N-terminal end of Nup98-FG. The binding of Tau to the Nup98-FG layer leads to a change in the refractive index of the medium close to the sensor surface, with the response signal, RU, being proportional to the bound mass. Monitoring RU during the association and dissociation process allows us to extract the association constant, k_on_, and the dissociation constant, k_off_ (Figure 1E). The equilibrium dissociation constant, K_D_, can be derived from k_on_ and k_off_ (Appendix A) and characterizes the overall binding affinity. Of note, R_max_ depends on the binding capacity for the analyte, which is derived from the mean grafting distance of the ligand (Appendix A). Because variations in grafting distance during Nup98-FG immobilization lead to differences in overall R_max_ values between experiments, binding responses could only be compared directly on the same Nup98-FG layer within one experiment.

To examine the interaction of recombinant human full-length Tau (2N4R produced in *E. coli*, thus devoid of PTMs) with Nup98-FG, increasing concentrations of Tau (0.078–4 μM in PBS) (Figure 1F), were applied to surface-tethered Nup98-FG layers. This led to an increase in RU, indicating the binding of Tau to the Nup98-FG layer, and a K_D_ of 0.4 μM was extracted from the Tau concentration-dependent responses, RU (Figure 1F,G). No RU increase was detected for the inert reference molecule PUT3. Interestingly, the binding of Tau to Nup98-FG/FS, a mutant in which all phenylalanine residues were replaced by serine residues, and which therefore lacked the FG-repeats essential for NTR binding [33,34,35], showed comparable K_D_ and R_max_ values (Appendix A). This indicated that the presence of phenylalanine residues was not important for the interaction of Tau with Nup98-FG.

### 2.3. The Tau Repeat Domain Mediates the Binding to Nup98

To investigate which part of Tau mediates the interaction with Nup98-FG layers, we tested the binding of different recombinant Tau variants and constructs. For this, we mostly employed BLI with Nup98-FG layers immobilized on the BLI tips, similar to SPR, because it allowed us to compare the binding of different Tau variants with higher throughput.

Tau exists in different splice isoforms that carry 0, 1, or 2 negatively charged N-terminal inserts (0N, 1N, 2N isoforms; Figure 2A and Appendix A). Since many of Tau’s interactions, including the binding to MTs, is driven by electrostatic interactions and involves charged protein domains, we investigated whether the presence of the negatively charged inserts would make a difference for the binding of Tau to Nup98-FG. When measuring the binding of Tau variants 2N4R (net charge at pH 7.0 = +3), 1N4R (net charge at pH 7.0 = +7), and 0N4R (net charge at pH 7.0 = +14) to Nup98-FG in BLI, we did not detect differences in the binding behavior leading to similar k_on_, k_off_, and K_D_ values (Figure 2B,C). These data showed that the presence of negatively charged domains in the N-terminal half of Tau, and hence the net charge of Tau, did not influence its binding to Nup98-FG. These findings suggested that the positively charged repeat domain of Tau may drive the interaction. To test this hypothesis, we compared the binding of full-length Tau with that of the repeat domain (TauRD, aa 244–372, net charge at pH 7.0 = +10.4), the N-terminal half (N-term, aa 1–257, net charge at pH 7.0 = −5.6; Figure 2A), and PBS controls. TauRD (K_D_ = 0.54 μM) showed binding to Nup98-FG with an even higher affinity than full-length Tau (K_D_ = 0.86 μM), whereas the binding of the N-term (K_D_ = 1.25 μM) was low compared to full-length Tau (Figure 2D,E,F and Appendix A). The comparably strong binding of TauRD could be due to the absence of the unstructured N-terminal half, which has previously been implicated in shielding interactions of the Tau repeat domain. Tau can adopt a ‘paper clip’ fold in solution [36], in which the unstructured N-terminal half folds back onto the repeat domain. This folding of the N-terminus is thought to shield interactions of the repeat domain, i.e., leading to a slower aggregation into paired helical filaments of full-length Tau compared to TauRD [37]. Interestingly, when investigating the association and dissociation signal of Tau in more detail, we found that after the dissociation phase the RU signal did not go back to the base-line level prior to the binding of Tau (Figure 2D, Δresponse). TauRD showed a similar effect, suggesting that Tau and TauRD accumulated on the Nup98-FG layer. The accumulation of Tau was less pronounced in experiments with less densely grafted Nup98-FG layer (=smaller RU values, e.g., in Figure 2C).

In summary, we found that the positively charged Tau repeat domain was sufficient to allow binding of Tau to Nup98-FG and that Tau accumulated on immobilized Nup98-FG layers.

### 2.4. Phosphorylation Increases the Binding and Accumulation of Tau to Nup98-FG

In human AD brains, we found that phosphorylated Tau accumulated around neuronal nuclei (Figure 1D), suggesting that phosphorylation may also increase the binding of Tau to FG-Nups in NPCs. Indeed, we previously found that recombinant phosphorylated Tau (P-Tau) strongly binds to Nup98-FG by SPR [12]. We confirmed these data in the current study by analyzing the SPR binding of phosphorylated Tau (P-Tau) purified from insect cells (8 ± 5 phosphates per Tau molecule [38]; net charge at pH 7.0~−9) to Nup98-FG layers in vitro. Compared to non-phosphorylated Tau (from *E. coli*), P-Tau showed a different binding behavior with a strong increase in R_max_ and a ~10-fold higher K_D_ (Tau: K_D_ = 1.13 μM, R_max_ = 351 RU; P-Tau: K_D_ = 12.2 μM, R_max_ = 1498 RU; Figure 3A,B). However, kinetic analysis of the association and dissociation rate constants revealed that the binding of Tau was characterized by a fast association (k_on_~1 × 10^4^ to 1 × 10^6^ M^−1^ s^−1^) and dissociation (k_off_~1 × 10^−2^ to 1 × 10^2^ s^−1^; Figure 3C), indicating transient binding. The kinetic maps showed two binding behaviors observed for Tau: (1) slow binding and dissociation, and (2) a more transiently bound fraction. In contrast, P-Tau showed slower association (k_on_~1 × 10^2^ M^−1^ s^−1^) and dissociation rates (k_off_~1 × 10^−4^) compared to Tau. This confirms our hypothesis that the additional negative net charge enhanced the interaction of P-Tau with the Nup98-FG layer, which had an overall positive net charge of +7.1 at pH 7.0. The use of non-interacting BSA (spikes in SPR profile; see Figure 1F and Figure 3A) further allowed us to measure the corresponding in situ height changes in the Nup98-FG layer [39]. Indeed, increasing concentrations of P-Tau caused a significantly larger increase in layer height than Tau (Figure 3D), indicating that P-Tau was irreversibly ‘deposited’ on the Nup98-FG layer. A similar effect was observed based on BLI (Appendix A). To test whether the deposition of P-Tau involved Nup98 FG-repeats, we repeated the experiments on layers of Nup98-FG/FS. However, the binding kinetics, K_D_, and maximal binding response, R_max_, of P-Tau were similar for wildtype Nup98-FG and Nup98-FG/FS (Appendix A), indicating that other residues and different non-FG interactions play a role in driving the accumulation of P-Tau within the FG-Nup layers. Phosphorylation of serine and threonine residues added a substantial amount of negative charges (average of −1.5 per phosphate group) to Tau, whereby most phosphorylation sites were located adjacent to the repeat domain in the proline-rich domains P1 and P2, and in R’. Phosphorylation in these Tau domains is associated with somatodendritic missorting and decreased MT binding of Tau [23].

Acetylation is another PTM that influences the biological [40] and condensation [41] activities of Tau. Acetylation reduces the net charge of Tau by removing positive charges on lysine side chains at different positions in Tau. To test whether acetylation would change the binding of Tau to Nup98-FG, we performed SPR experiments with Tau that was in vitro acetylated using the acetyl transferases P300 and CBP (Appendix A) [41]. Notably, acetylated Tau (ac-Tau) showed binding behavior similar to Tau and no layer height increase (Appendix A).

While phosphorylation and acetylation both led to a reduction of Tau’s net charge (phosphorylation added extra negative charges, whereas acetylation removed positive charges), only phosphorylation strongly affected the binding of Tau to Nup98-FG.

To summarize, our results show that Tau exhibited rapid and transient interactions with Nup98-FG, whereas the slow association of P-Tau facilitated its accumulation with Nup98-FG (Figure 3E).

### 2.5. Oligomerization Reduces the Binding of Tau to Nup98-FG

In the context of AD and tauopathies, Tau phosphorylation is linked to pathological oligomerization, which is thought to precede Tau aggregation into fibrillar aggregates [42]. Multiple studies indicated that Tau oligomers can induce neurotoxicity even in the absence of NFT formation [43]. Whether oligomers of phosphorylated Tau are involved in Tau-induced NCT impairment is not known. To investigate how the oligomerization of P-Tau impacts its binding to FG-Nups, we produced a phosphorylated pro-aggregation FTD-mutant TauΔK280 in insect cells. For both P-Tau and P-TauΔK280, we tested the binding to Nup98-FG layers in their monomeric (freshly thawed protein) and oligomeric (incubated at room temperature for 48 h) states. Right before the BLI experiments, we confirmed the assembly state of Tau in these preparations by dynamic light scattering (DLS). Fresh protein samples all contained mostly Tau monomers (radius ~10 nm) and some small clusters (radius ~150 to 300 nm). After incubation at room temperature, both P-Tau and P-TauΔK280 contained almost exclusively Tau oligomers (radius ~600 nm), whereas non-phosphorylated wildtype Tau did not change composition (Figure 4A and Appendix A). Surprisingly, when comparing the binding of monomeric versus oligomeric P-Tau and P-TauΔK280 to Nup98-FG, we found a significantly reduced binding for P-Tau and P-TauΔK280 oligomers (Figure 4B,C). This indicated that soluble Tau oligomers are less likely to interact and accumulate with Nup98-FG, possibly because the surface of Tau oligomers and/or their size prevent these interactions.

## 3. Discussion

Tau is an intrinsically disordered axonal protein. Under neuronal stress conditions and neurodegenerative diseases such as AD, Tau becomes missorted into the neuronal soma [6], where interactions with the nuclear envelope and NPCs can take place. In fact, the colocalization of Tau with NPCs was shown in AD brains [12] and in a cell model of Tau aggregation [13], where a direct interaction of Tau with Nup98 has been suggested. Furthermore, Tau was shown to inhibit the nuclear transport of proteins and RNA through NPCs in models with somatodendritic Tau accumulation and in AD brains [12,21]. Nuclear transport relies on proper function and permeability of FG-Nup filled nuclear pores, and clogging of the pores, for example, by binding of P-Tau, can explain the nuclear transport deficits observed upon missorting of Tau.

We investigated whether Tau can directly interact with FG-rich nucleoporins. Using surface plasmon resonance and biolayer interferometry, we were able to describe the binding of different Tau variants, PTM states, and assembly forms to immobilized layers of the disordered Nup98 N-terminal half, Nup98-FG. Unlike Nup98-FG in solution, surface-tethered Nup98-FG polypeptide chains presented on a SPR chip can mimic their C-terminal attachments to the inner surface of the pore ring. In summary, our data show different interaction scenarios for different Tau forms (Figure 5): (1) a transient binding of (unmodified) Tau to Nup98-FG layers is mediated by the Tau repeat domain, (2) a slow association and deposition of phosphorylated Tau (P-Tau) on Nup98-FG, and (3) a hindered interaction of pre-formed Tau oligomers.

For Tau-induced deficits of NPC-mediated transport, these findings offer an interesting and somewhat unexpected interpretation. In neurons, most Tau is to some extent phosphorylated, especially when it is released from axonal microtubules and sent into the neuronal cell body. Phospho-Tau therefore is the relevant Tau species at NPCs. Our data showing the pronounced accumulation of P-Tau on Nup98-FG layers may thus explain how phospho-Tau deposits clog NPCs in neurons.

Phosphorylation is the major PTM and regulates the interactions of Tau. It introduces negative charges along the polypeptide sequence and thereby changes the electrostatics of the protein. Phosphorylation can occur at ~80 different sites in the 2N4R isoform of the protein. P-Tau (2N4R) used in this study is similar to AD Tau with regard to its phosphorylation status: both P-Tau and AD Tau contain a heterogenous mix of phospho-Tau molecules, and most phospho-sites are located in the proline-rich domains, P1 and P2, and in the region C-terminal to the repeat domain [38,44,45]. Phospho-Tau has a low net charge (net charge at pH 7: −9) compared to non-phosphorylated Tau (net charge at pH 7: ~3), which may contribute to the accumulation of Tau on Nup98-FG layers. Acetylation, which does not introduce negative charges but neutralizes positively charged lysine side chains, mostly in the Tau repeat domain and near the N-terminal inserts, did not affect the binding of Tau to Nup98-FG. Hence, we suggest that the presence and distribution of negative charges in P-Tau could be important for Tau’s accumulation on Nup98-FG layers.

Comparing different Tau domains, we found that a positive net charge correlated with the interaction of Tau with Nup98-FG (binding of TauRD (net charge +10.4) > 2N4R (+3.1) > N-term (−5.6)). These findings suggest that the interactions of Tau with Nup98-FG may be driven by electrostatic interactions of the positively charged repeat domain. The charge of the repeat domain was also influenced by the presence or absence of the alternatively spliced pseudo-repeat 2 (R2) in 3R versus 4R Tau isoforms. It remains to be investigated to which extent the binding of Tau to Nup98-FG is influenced by R2. Notably, 3R and 4R Tau isoforms are expressed at similar levels in adult human brains. In the Nup98-FG sequence, the only negatively charged domain available for electrostatic interactions with the Tau repeat domain is the GLEBS domain, a binding domain important for mRNA export [46,47]. Whether this particular sequence is important for the binding to Tau needs to be addressed further. However, as a consequence of TauRD:GLEBS interactions, mRNA export and thus translation could be affected. Keeping the interaction of Tau with Nup98-FG transient could thus be important for maintaining mRNA transport.

Disease-associated Tau phosphorylation, however, seems to hinder a fast interaction but does not fully prevent it. P-Tau binding to Nup98-FG layers is characterized by a slow association, probably hindered by the additional negative charges flanking the repeat domain. Once bound, P-Tau strongly accumulates on the Nup98-FG layer, which could be caused by increased Tau–Tau interactions. In cells, phospho-Tau accumulation on FG-Nups in NPCs could clog the nuclear pore and thereby could impair nucleocytoplasmic transport. This could explain the neuronal nuclear transport defects that have been observed in human AD brain and mutant Tau transgenic mouse and fly models of tauopathy [12,21].

In the cascade of molecular Tau changes associated with neuronal stress and toxicity, phosphorylation and missorting are followed by oligomerization, accumulation, and eventually the aggregation into fibrillar forms of Tau [48,49]. The overall physicochemical properties largely differ between Tau monomers and oligomers (e.g., accessible surface charge and size), which likely has a large impact on the interaction with Nup98-FG—as it does for all other interactions of Tau. Importantly, although expecting an enhanced interaction, we found that oligomerized phospho-Tau lost the capability of binding to Nup98-FG layers. This suggests that the interaction of Tau with the FG-Nups, and the still speculative consequences, occur early in the Tau toxicity cascade, namely, prior to oligomer formation. In fact, the localized strong accumulation of P-Tau at the nuclear membrane could trigger Tau aggregation, which occurs spontaneously out of local high (~20–30 mM) concentrations of Tau. This has been observed in vitro for Tau in solution, as well as for biomolecular Tau condensates [50]. In a hypothetical scenario, P-Tau accumulation with FG-Nups could even induce oligomerization and subsequent pathological Tau aggregation. Oligomerization changes the conformation of included Tau molecules and may make Tau motifs needed for the interaction with Nup98-FG inaccessible.

In AD brains, we observed Tau accumulating in a layer around the nucleus only in neurons with diffuse cytosolic P-Tau, i.e., not in neurons with aggregated Tau. However, we previously showed that Nup98 and other FG-Nups co-aggregate with Tau and accumulate in NFTs of AD brains [12]. Since Nup98 is involved in many cellular processes, including nuclear import and export, mitotic progression, and regulation of gene expression, a removal of Nup98 from the NPC could have severe consequences.

In summary, our data provide important details about the interactions between Tau and Nups embedded in NPCs and help to explain the impact of stress- and disease-associated Tau PTMs, i.e., phosphorylation and truncation of Tau domains, for this interaction. Our findings support a model, in which somatodendritic phospho-Tau in stressed neurons accumulates with FG-Nups in NPCs, where it could hinder nuclear transport. This interaction can occur already prior to Tau oligomerization and aggregation. It needs to be further clarified whether the interaction of phospho-Tau with FG-Nups has a functional reason or is a pathological artifact of P-Tau sorting in the somatodendritic compartment.

## 4. Materials and Methods

### 4.1. Human Samples

Human AD (Braak 4/5) and age-matched control brain samples were received from the Biobanks at the University Tübingen and the Charité, Berlin, Germany. For nuclei isolation, frontal cortex from a control (Braak 1/2, 77 year, male) and an AD (Braak 5/6, 77 year, female) were used. For Immunolabeling of P-Tau, three AD cases (Braak 4–5, mixed gender) were used.

### 4.2. Tau Detection in AD Brain Sections (IHC)

Paraffin brain sections of hippocampal and entorhinal cortex were immunofluorescently labeled for phospho-Tau (pS396/pS404, PHF1 epitope, and pT231, AT180 epitope) as previously described [12]. Sections were imaged in epifluorescence using 20× air and 63× oil objectives under a widefield fluorescence microscope (Eclipse-Ti, Nikon, Minato-ku, Japan).

### 4.3. Expression and Purification of Recombinant Proteins (E. coli and SF9 Tau)

All plasmids were verified by Sanger sequencing prior to protein production. Human Tau isoforms and variants (2N4R, 1N4R, 0N4R, TauRD, N-term) were expressed in *E. coli* BL21 Star (DE3) (Invitrogen, Waltham, MA, USA) as previously described [37]. Protein expression was induced with 0.5 mM IPTG at OD600 = 0.6 for ~3 h at 37 °C. Cells were harvested, resuspended in lysis buffer (20 mM MES, 1 mM EGTA, 0.2 mM MgCl_2_, 1 mM PMSF, 5 mM DTT, protease inhibitors (Pierce Protease Inhibitor Mini Tablets, EDTA-free), and lysed using a French press. After initial purification by adding 500 mM NaCl and boiling at 95 °C for 20 min, cell debris was removed by centrifugation and the supernatant was dialyzed against a low salt buffer (Buffer A: 20 mM MES, 50 mM NaCl, 1 mM MgCl_2_, 1 mM EGTA, 2 mM DTT, 0.1 mM PMSF, pH 6.8), filtered (0.22 µm membrane filter), run through a cation exchange column (HiTrap SP HP, 5 mL, GE Healthcare, Chicago, IL, USA), and eluted with a high salt buffer (Buffer B: 20 mM MES, 1000 mM NaCL, 1 mM MgCl_2_, 1 mM EGTA, 2 mM DTT, 0.1 mM PMSF, pH 6.8). Fractions containing Tau were pooled, concentrated using spin column concentrators (Pierce Protein concentrators; 10–30 kDa MWCO, Thermo Fischer Scientific, Waltham, MA, USA), and run through a size exclusion column (Superose 6 10/300, GE Healthcare, Chicago, IL, USA). Fractions containing purified monomeric Tau were concentrated as before and buffer exchanged to PBS, 1 mM DTT, pH 7.4. The final Tau concentration was measured with BCA assay (BCA kit, PIERCE), and the protein was stored at −80 °C.

Phosphorylated Tau (P-Tau and P-TauΔK280) was expressed in insect (SF9) cells by infection with a recombinant baculovirus containing a pTriEx-Tau plasmid that encodes human full-length Tau (2N4R or TauΔK280) with a N-terminal Strep-Tag [51]. After cell lysis, the cleared lysate was applied to a 10 mL Strep-Tactin column (Iba) and a gel filtration column (Superdex200 16,600, GE Healthcare, Chicago, IL, USA). The purified protein was stored in PBS containing 1 mM TCEP at −80 °C.

Nup98-FG and Nup98-FG/FS plasmid constructs were bought from GenScript (Piscataway, NJ, USA) and expressed and purified as described before [52]. Briefly, Nup98-FG represents the N-terminal FG-domain of Nup98 (aa 1-498), with one additional cysteine at the N-terminus to facilitate covalent binding to the gold surface of the SPR chip; Nup98-FG/FS is the full F/S mutant of Nup98-FG.

### 4.4. Tau in Vitro Acetylation

Recombinant human Tau was acetylated using CBP (BML-SE452; Enzo) and P300 (BML-SE451; Enzo) acetyl transferases. The reaction was performed by mixing reaction components up to the following final concentrations: 200 µM Tau, 0.028 mg/mL CBP or P300, 20 mM Ac-CoA, 0.5 mM PMSF and 5 mM EGTA in 25 mM HEPES, pH 7.4, containing 100 mM KCl, 5 mM MgCl_2_ and 1 mM TCEP. The reaction mixture was incubated overnight at 30 °C on a shaker (350 rpm). The samples were then boiled for 20 min at 95 °C to inactivate the enzymes. After 30 min centrifugation at 4 °C and maximum speed, the supernatant was collected, aliquoted, and either used immediately or flash-frozen for further use. The level of acetylation was determined by mass spectrometry, which confirmed the presence of 6–8 acetylated groups in both CBP and P300 acetylated samples.

### 4.5. Surface Plasmon Resonance

SPR measurements were performed using a Biocore T200 (GE Healthcare, Life Sciences, Piscataway, NJ, USA). The thiol-modified Nup98-FG-SH protein and the inert control polymer PUT3 (C_17_H_36_O_4_S, hydroxyl-terminated tri-ethylene glycol undecane thiol, HS-(CH_2_)-(OCH_2_CH_2_)_3_-OH) were immobilized to the gold surface of the sensor chip (SIA kit; GE Healthcare, Life Science, Piscataway, NJ, USA). Tau proteins were then injected in increasing concentrations between 7.6 nM to 4 μM, and differences of the Nup98-FG layer height were determined. BSA was injected after each Tau injection as described previously [36]. The equilibrium binding constants were calculated using one- or two-component Langmuir fits [53]. Sensograms presented in one graph were obtained from measurements using the same gold sensor chip and the same immobilization of Nups. A detailed protocol is provided by Diez et al. [54].

### 4.6. Bio-Layer Interferomertry

Bio-layer interferometry experiments were performed on a BLItz instrument (ForteBio, Inc., Fremont, CA, USA) with BLItz Prot analysis software. His-tagged Nup98-FG (1 mM) was immobilized on commercially available ForteBio Ni–NTA or HIS biosensors. The provided ‘Dip and Read’ assay was used to monitor specific protein binding. Binding of the specific protein to the surface leads to the increase in the optical thickness of the bio-layer, resulting in the shift of the wavelength (in nm). All experiments were performed in PBS.

### 4.7. Dynamic Light Scattering (DLS) Measurements

The hydrodynamic radius distribution of Tau protein samples was measured using dynamic light scattering (DLS) analysis (Zetasizer nano ZSP, Malvern Instruments, Malvern, UK). All size measurements were done in triplicates at 4 µM in PBS.

### 4.8. Human Brain Nuclei Preparation

Frozen brain tissue was thawed on ice, and ~1 g piece of human tissue (frontal cortex) was placed into 4 mL of homogenization buffer (0.32 M sucrose, 5 mM Hepes, pH 7.4) in a 5 mL glass vessel. Homogenization was performed using 12 strokes at 900 rpm (Sartorius Potter S) on ice; 0.5 mL of brain homogenate was placed into a fresh 1.5 mL Eppendorf tube as total homogenate. The remaining 3.5 mL was centrifuged at 1000× *g* for 10 min at 4 °C to separate the cytosolic fraction (supernatant) from the nuclei and cell debris (pellet). The pellet was resuspended in 3.5 mL lysis buffer (0.32 M sucrose, 5 mM CaCl_2_, 3 mM MgAc_2_, 0.1 mM EDTA, 10 mM Tris-HCl pH 8.0, freshly made 1 mM DTT, and 0.1% Triton X-100) and placed into a pre-chilled ultracentrifuge tube (Beckman Coulter); 7.5 mL of a sucrose solution (1.8 M sucrose, 3.0 mM MgAc_2_, 1 mM DTT and 10 mM Tris-HCl, pH 8.0) was pipetted at the bottom of the ultracentrifuge tube containing the homogenized sample to create a concentration gradient with the homogenate on top of the sucrose solution. The tube weight balance was adjusted by adding lysis buffer to the top layer. Samples were ultra-centrifuged at 107,000× *g* (25,000 rpm) using an SW41 Ti rotor for 2.5 h at 4 °C. After centrifugation, the supernatant was carefully removed, along with the layer of debris. The nuclei pellet remained at the bottom of the tube and was carefully resuspended in 0.5 mL of chilled homogenization buffer. To estimate the protein concentration of the isolated nuclei, a small fraction was diluted in RIPA buffer (Sigma, R0278; 150 mM NaCl, 1.0 % IGEPAL^®^ CA-630, 0.5 % Natriumdesoxycholat, 0.1 % SDS, 50 mM Tris, pH 8.0).

### 4.9. Coomassie, SDS-PAGE, and Western Blot

For SDS-PAGE, protein samples were mixed with 6× reducing SDS-containing loading buffer, boiled at 95 °C for 5 min, and separated by SDS-PAGE (NuPAGE 4–12% Bis-Tris, Invitrogen) using MES buffer. From recombinant protein, 0.5 µg protein was loaded per well and Coomassie stained. From brain nuclei samples, 10 µg of protein was loaded per well and blotted on a nitrocellulose membrane (Amersham) for western blot analysis. The membrane was blocked with 3% BSA in PBS containing 0.05% Tween (BSA/PBS-T) for 1 h, followed by incubation with primary antibodies in blocking buffer overnight at 4 °C (primary antibodies: rabbit anti-phospho-Tau (PHF1 = epitope pS396/pS404; Tau pS396 (1:10,000; Abcam)/Tau pS404 (1:1000, Cell Signaling); rabbit anti-Histone 3 (H3; 1:2000, Sigma). After washing three times for 5 min with BSA/PBST, incubation with HRP (horse-radish-peroxidase)-coupled secondary antibodies followed (anti-rabbit or anti-mouse HRP, 1:2000, Sigma) for 1 h at room temperature. After incubation, the membrane was washed again 3× with BSA/PBS-T, and detection of secondary antibody was performed by visualizing chemiluminescence after the addition of LumiGLO/Peroxide (Cell Signaling) with an image reader (Fusion SL system, Vilber Lourmat).

## Figures and Tables

**Figure 1 ijms-23-03495-f001:**
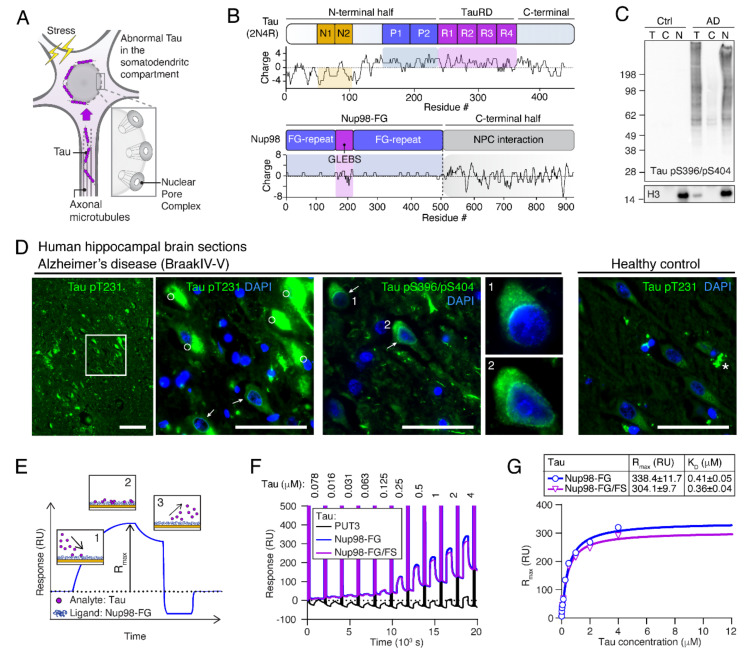
Tau accumulates at neuronal NEs in AD brains and binds to Nup98-FG. (**A**) Scheme of cellular Tau localization. Under normal conditions, the microtubule-associated Tau protein is restricted to the neuronal axon. Upon stress and under disease conditions, Tau mislocalizes to the neuronal cell body where it comes into close proximity with the nucleus; interactions with components of the NPC are enabled. (**B**) Domain structure and charge distribution along the sequence of human full-length wild-type Tau (2N4R, 441 aa) and Nup98 (937 aa) analyzed using EMBOSS (http://www.bioinformatics.nl/cgi-bin/emboss/charge; accessed on 10 January 2022) with a sliding window of 8 aa. The N-terminal inserts (N1, N2; yellow) are predominantly negatively charged, the pseudo-repeat domain (R1-R4; purple) and adjacent Proline-rich domains (P1, P2; blue) are positively charged. For Nup98, the C-terminal part (grey) is anchored in the NPC scaffold, whereas the N-terminal part (Nup98-FG; blue/purple) protrudes unstructured into the pore of NPC and contains two non-charged FG-rich domains separated by the negatively charged GLEBS domain (Gle2-binding sequence). (**C**) Western Blot analysis of human postmortem brain tissue shows phosphorylated Tau (pS396/pS404) in isolated nuclei (N) of AD but not control cases (T = Total lysate; C = Cytosol; N = Nuclei). Histone H3 is loaded to control for nuclei enrichment. (**D**) Immunolabeling of phosphorylated Tau (pT231 and pS396/pS404) in AD Braak 4/5 hippocampal brain sections shows the side-by-side existence of neurofibrillary tangles (NFTs, white circles) filled with aggregated phospho-Tau and neurons with phospho-Tau accumulating in a layer at the nuclear envelope (white arrows). Notably, compared to neuronal nuclei, cell nuclei of glia cells appear smaller and brighter in the DAPI channel. The asterisk in the healthy control brain section indicates autofluorescence of lipofuscin. Scale bars are 50 µm. (**E**) Schematic SPR sensogram showing a plot of resonance signal (RU) versus time. After immobilization of Nup98-FG ligands on the sensor chip, the continuous injection of the Tau analyte sample enables the binding of Tau to the Nup98-FG layer and an increase in RU in the association phase (1). A steady-state occurs when binding and dissociating molecules are in equilibrium and RU reached a maximum, RU_max_ (2). Upon changing the flow medium to PBS, Tau dissociates from the Nup98-FG layer and RU decreases (3). (**F**) In SPR response curves, sensograms at increasing Tau concentrations are recorded, whereby the association and dissociation of Tau on an Nup98-FG layer is recorded. After each applied Tau concentration, flushing the chip with bovine serum albumin (BSA; inert control not binding to Nup98-FG) induces a spike in the profile. At the end of each spike, the thickness of the Nup98-FG layer is measured before applying the next Tau concentration. Tau (2N4R) shows comparable binding to Nup98-FG as well as to the Nup98-FG/FS mutant. (**G**) Equilibrium fit (Langmuir binding isotherm) to maximal RU (RU_max_) versus Tau concentration for the binding reactions shown in (**F**). R_max_ and binding constants (K_D_) are shown in the table as the value ± SD.

**Figure 2 ijms-23-03495-f002:**
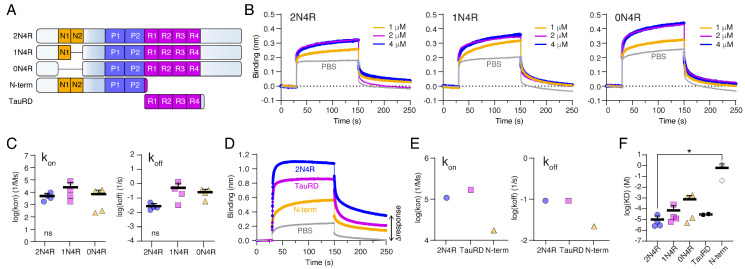
The Tau repeat domain mediates the binding to Nup98-FG. (**A**) Domain structure of Tau 2N4R, 1N4R, and 0N4R isoforms as well as the N-terminal (N-term; aa 1–257) and the Tau repeat domain (TauRD, aa 244–372) constructs. (**B**) Representative BLI measurements of Tau isoforms 2N4R, 1N4R, and 0N4R measured at 1, 2, and 4 µM binding to an immobilized Nup98-FG layer. Reference measurements for PBS were recorded for each construct individually. (**C**) Association constants (k_on_) and dissociation constants (k_off_) for Tau isoforms binding to Nup98-FG. Data shown as the mean + SD from four replicates. One-way ANOVA with Tukey’s post-test. (**D**) Representative BLI measurements of 2N4R, TauRD, and N-term binding to Nup98-FG. The PBS reference is shown as well. An increase in the Nup98-FG layer height (Δresponse) results from incomplete detachment of Tau during the dissociation phase. (**E**) Association (k_on_) and dissociation values (k_off_) from (**D**). (**F**) Summary of equilibrium constants (K_D_) for 2N4R, 1N4R, 0N4R, TauRD, and N-terminal Tau variants. Data are shown as the mean + SD. One-way ANOVA, Dunnett’s test for multiple comparison, 2N4R versus N-term: * *p* = 0.0480 (* *p* < 0.05).

**Figure 3 ijms-23-03495-f003:**
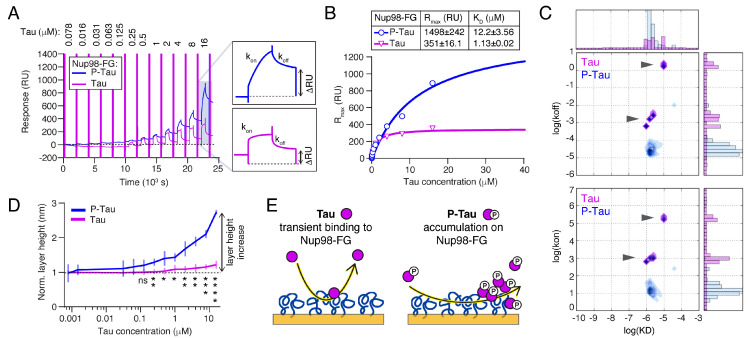
Phosphorylation leads to accumulation of Tau within Nup98-FG layers. (**A**) SPR response curves for Tau and P-Tau to Nup98-FG. Illustrated zoom-in highlights the difference in association and dissociation, showing slower k_on_ and k_off_ and a larger R_max_ and ΔRU for P-Tau compared to Tau. (**B**) Equilibrium fit (Langmuir binding isotherm) to RU_max_ plotted versus Tau concentration for the binding reactions shown in (**A**). In the table, fit results are shown as the value ± SD. (**C**) Kinetic maps for Tau (purple) and P-Tau (blue) binding to Nup98-FG show log (k_off_) (top) and log (k_on_) (bottom) plotted versus log (K_D_). Notably, two groups of k_on_ and k_off_ are detected for Tau (black arrow heads). (**D**) Layer height increase (derived from the differences in the BSA injection signals in the sample versus reference channels) of Tau and P-Tau shows accumulation of P-Tau on Nup98-FG layer starting at 0.25 µM p-Tau. Data are shown as the mean + SD. Comparison of Tau and P-Tau at individual Tau concentrations with unpaired Student’s *t*-test, 0.25 µM: *p* = 0.0077; 16 µM: *p* < 0.0001 (* *p* < 0.05, ** *p* < 0.01, *** *p* < 0.001, **** *p* < 0.0001). (**E**) Model of different binding behaviors of Tau and P-Tau to immobilized Nup98-FG layers.

**Figure 4 ijms-23-03495-f004:**
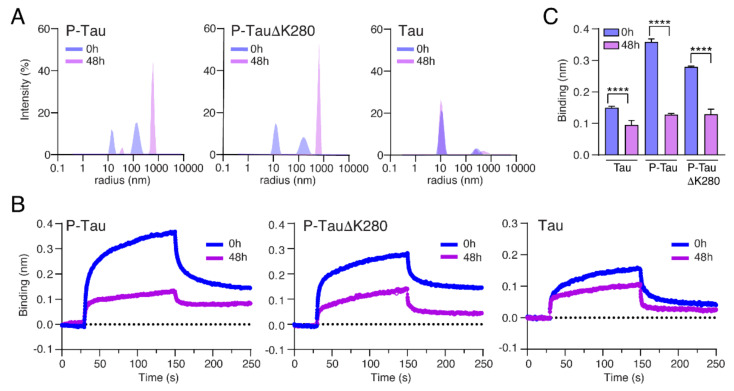
Oligomerization reduces the binding of Tau to Nup98-FG. (**A**) DLS size distribution profiles for Tau, P-Tau, and P-TauΔK280 at 0 h (blue) and 48 h (purple) of the oligomerization process (spontaneous oligomerization at room temperature). Both P-Tau and pro-aggregant P-TauΔK280, but not Tau, show a shift from Tau monomers (radius ~15 nm) at 0 h towards oligomers (radius ~600 nm) at 48 h. (**B**) BLI measurements show a pronounced reduction in binding for oligomerized 48 h-old P-Tau and P-TauΔK280. Tau showed less reduction in binding at 48 h. (**C**) Quantification of the binding (nm) of monomeric (0 h) and oligomeric (48 h) Tau versions at the end of the association phase (*t* = 150 s). Data are the mean + SD of the last five values of the association phase. Two-tailed Student’s *t*-test, unpaired, **** *p* < 0.0001.

**Figure 5 ijms-23-03495-f005:**
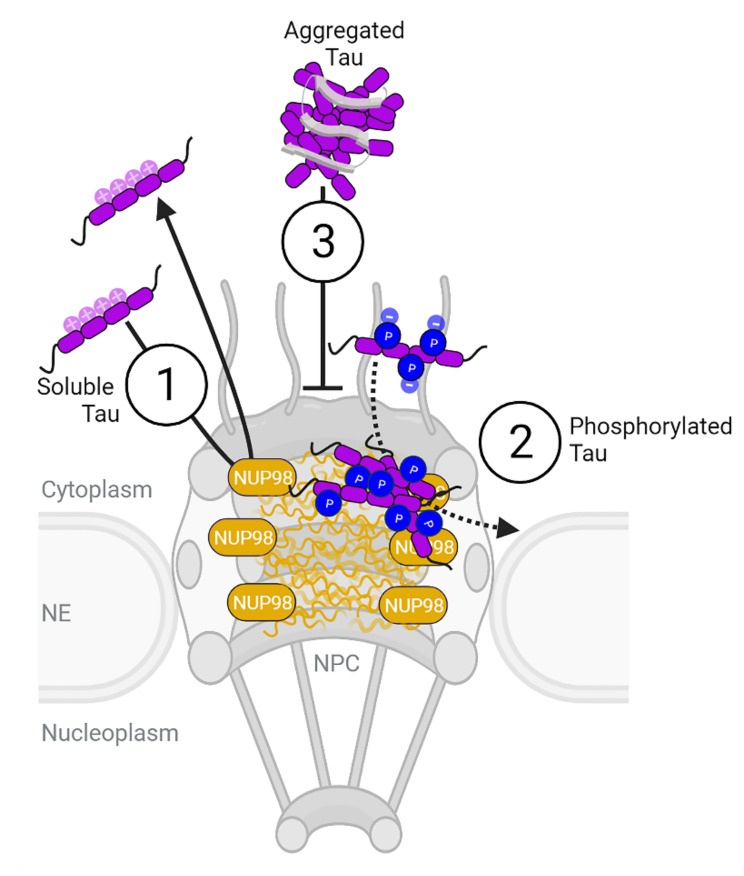
Potential interactions of Tau with Nup98-FG in NPCs. When coming in proximity of the nuclear envelope, e.g., upon stress-induced missorting into the neuronal somatodendritic compartment, Tau can come in contact with NPCs. Here, Tau can interact with the disordered peptide chains of FG-Nups, including Nup98-FG, that form the permeability barrier inside the pore of the NPC. Thereby, non-phosphorylated Tau (1) interacts transiently and reversibly with the FG-Nups. In contrast, phosphorylated P-Tau (2) binds with lower affinity but more stably, which leads to its accumulation with Nup98-FG. Importantly, most—if not all—Tau close to the nucleus of neurons is phosphorylated and, thus, can accumulate with the FG-Nups. Similar observations were made in postmortem human AD brains, where phosphorylated Tau forms a layer around nuclei at the nuclear envelope. Whether P-Tau accumulation with the FG-Nups has a functional purpose or is a stress/disease-associated pathological consequence of Tau missorting remains to be investigated. Tau oligomers (3), however, do not bind or accumulate with Nup98-FG, suggesting that oligomerization and aggregation are not essential for Tau accumulation on neuronal NPCs in AD (created with BioRender).

## Data Availability

All data are contained within the article or Appendix A.

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
