# Peer review of "Phosphorylation but Not Oligomerization Drives the Accumulation of Tau with Nucleoporin Nup98"

_ijms, 2022, doi:10.3390/ijms23073495_

Round 1

Reviewer 1 Report

The authors further characterized the factors influencing the interaction between tau and Nup98. The in vitro studies seem to be sound and demonstrates new findings. I have some minor comments regarding writings and experiments using human postmortem tissue.

Abstract: I believe NCT should be spelled out at first appearance also in the abstract.

Line 37: Although excitatory neurons are reported to be more prone to tau aggregation, I’m not sure if tau aggregation is limited to excitatory neurons.

Line 106-108: Although I agree that Figure 1(A-)D suggest that Tau enriches at outer nuclear layer in some cells of AD brain, I’m not sure if these cells are definitely neurons by the presented image. It would also be better if we could know how prevalent these findings are in AD brains.

Braak stage 5-6 AD brains should contain tau aggregates and Figure 1C suggests the presence of high molecular weight tau species. I couldn’t find results demonstrating that tau enrichment at outer nuclear layer occur “prior to tau aggregation” at this point (using human postmortem tissue).

The phrase “~neuronal nuclei of neurons of AD brain” seems to be redundant.

Figure 1C: H3 Western blot data confirms that nuclear fraction did not contaminate into cytosolic fraction. Did you confirm that the tau bands in nuclear fraction are not due to contamination of large tau aggregates (similar to the size of nucleus)?

Results 2.3 The Tau “repeat domain” mediates the binding to Nup98

The results show that 4RD is sufficient for interaction with Nup98 and whether 3RD also interacts with Nup98 was not examined. I understand that the authors focused on the difference of N-terminal, but this point may better be mentioned at least once in the manuscript (perhaps in discussion).

Results 2.4 “Phosphorylation increases the binding and accumulation of Tau to Nup98-FG”

Could you kindly explain the difference from the previous report (Neuron 2018; 99(5): 925-940.e7.)? Since previous report already suggested that phosphorylation is important in tau-Nup98 interaction, the manuscript should clarify which are confirmatory experiments and focus on the difference in experimental design and new findings.

Discussion Line 401-402 “we observed Tau accumulating in a layer around the nucleus only in neurons with diffuse cytosolic P-Tau but not in neurons with aggregated Tau”

Was this shown in this manuscript or your previous report? Although the discussion is interesting, sufficient evidence should be included in the results section (or previous report should be referenced here).

Materials and Methods 4.1. 4.2.

If you used entorhinal cortex of Braak 5-6 AD brains, how did you differentiate between cells with tangles and those with pretangels only by immunostaining? If you really want to state that “tau enrichment at outer nuclear layer occur prior to tau aggregation” in human brain, again I don’t think the present Figure 1C,D is sufficient.

Author Response

Point by Point answer to Reviewer 1

The authors further characterized the factors influencing the interaction between tau and Nup98. The in vitro studies seem to be sound and demonstrates new findings. I have some minor comments regarding writings and experiments using human postmortem tissue.

Abstract: I believe NCT should be spelled out at first appearance also in the abstract.

Authors: We changed the manuscript accordingly.

Line 37: Although excitatory neurons are reported to be more prone to tau aggregation, I’m not sure if tau aggregation is limited to excitatory neurons.

Authors: We thank the reviewer for bringing up this point and removed the specification “excitatory” from the text.

Line 106-108: Although I agree that Figure 1(A-)D suggest that Tau enriches at outer nuclear layer in some cells of AD brain, I’m not sure if these cells are definitely neurons by the presented image. It would also be better if we could know how prevalent these findings are in AD brains.

Authors: We agree with the reviewer and added additional representative images of hippocampal human AD brain sections. The new images, taken from AD Braak IV/V hippocampal sections, show that neurofibrillary tangles (NFTs; indicated with white circles) and neurons with Tau around the nucleus (indicated with white arrows) are both located in the same brain region adjacent to each other. Healthy control brains show neither NFTs nor Tau around the nucleus. The size and shape of the nuclei, as well as the cell morphology, clearly classifies the cells shown as neurons. Glia cells nuclei are smaller and brighter in the DAPI channel (higher DNA density). We now mention the size difference of neuronal versus glia cell nuclei in the figure legend.

Braak stage 5-6 AD brains should contain tau aggregates and Figure 1C suggests the presence of high molecular weight tau species. I couldn’t find results demonstrating that tau enrichment at outer nuclear layer occur “prior to tau aggregation” at this point (using human postmortem tissue).

Authors: This has in part been addressed in the comment above. We also removed the phrase ‘prior to tau aggregation’.

The phrase “~neuronal nuclei of neurons of AD brain” seems to be redundant.

Authors: We changed the manuscript accordingly.

Figure 1C: H3 Western blot data confirms that nuclear fraction did not contaminate into cytosolic fraction. Did you confirm that the tau bands in nuclear fraction are not due to contamination of large tau aggregates (similar to the size of nucleus)?

Authors: During the nuclei preparation, most large debris and myelin gets removed by sucrose gradient centrifugation. However, Tau aggregates could indeed still contaminate the nuclei preparation and therefor give a false positive signal for P-Tau association with nuclei. In fact, this is one of the reasons why we performed high resolution immunofluorescence microscopy of brain sections labeled for P-Tau, which clearly show that P-Tau does accumulate at neuronal nuclei, independent of Tau aggregation.

Results 2.3 The Tau “repeat domain” mediates the binding to Nup98

The results show that 4RD is sufficient for interaction with Nup98 and whether 3RD also interacts with Nup98 was not examined. I understand that the authors focused on the difference of N-terminal, but this point may better be mentioned at least once in the manuscript (perhaps in discussion).

Authors: We agree with the reviewer that the difference between 3R vs. 4R Tau is also a very interesting topic, especially taking into consideration that 3R and 4R isoforms are present to similar amounts in adult human brain (Wang and Mandelkow 2015). Although not experimentally addressed in this manuscript, we added a short discussion on the 3R/4R topic, as suggested by the reviewer.

Results 2.4 “Phosphorylation increases the binding and accumulation of Tau to Nup98-FG”

Could you kindly explain the difference from the previous report (Neuron 2018; 99(5): 925-940.e7.)? Since previous report already suggested that phosphorylation is important in tau-Nup98 interaction, the manuscript should clarify which are confirmatory experiments and focus on the difference in experimental design and new findings.

Authors: We thank the reviewer for addressing this point. We indeed were able to show previously that phosphorylated Tau interacts with Nup98 (Eftekharzadeh et al. Neuron 2018); based on these observation, we aimed to characterize the interaction of Tau with Nup98-FG in more detail. In the current manuscript, we put a special focus on 1) investigating Tau domains contributing to this interaction, and 2) the comparison on different Tau modifications relevant in Tau biology, i.e. phosphorylation, acetylation, and oligomerization. Additionally, we also addressed the impact of aggregation-prone FTD-mutations. To make clear what has been shown before, we now added a sentence clearly stating that the effect of phosphorylation has been seen before, and that we reproduced and extended these data to other modifications. (Lanes 225-227).

Discussion Line 401-402 “we observed Tau accumulating in a layer around the nucleus only in neurons with diffuse cytosolic P-Tau but not in neurons with aggregated Tau”. Was this shown in this manuscript or your previous report? Although the discussion is interesting, sufficient evidence should be included in the results section (or previous report should be referenced here). 

Authors: With the added images showing hippocampal neurons with Tau around the nucleus adjacent to neurons filled with aggregated Tau (new panel D in Fig. 1), we provide the evidence for this statement.

Materials and Methods 4.1. 4.2.

If you used entorhinal cortex of Braak 5-6 AD brains, how did you differentiate between cells with tangles and those with pretangels only by immunostaining? If you really want to state that “tau enrichment at outer nuclear layer occur prior to tau aggregation” in human brain, again I don’t think the present Figure 1C,D is sufficient.

Authors: We added images from the hippocampal area stained for p-Tau, in which one can clearly see the difference between neurons filled with aggregated Tau and neurons having diffuse pTau in their soma and accumulations of pTau on the NE. As mentioned above, we also removed the statement that NE Tau occurs prior to Tau aggregation since we cannot add a temporal dimension in postmortem fixed tissue. We are working on this problem in another study.

Reviewer 2 Report

Diez et al. studied the interaction of Tau with Nup98 in the context of tauopathy, like Alzheimer’s disease. Using SPR and BLI, the authors measured association and dissociation of various Tau fragments with the N-terminal Nup98-FG or the full length of Nup98-SG mutant peptide and found TauRD and the phosphorylated Tau (either full length or delta K280) efficiently interact with Nup98-FG whereas acetylated Tau or oligomerized Tau did not. The data support the authors’ conclusions, and I am supportive of the publication of the paper. However, I would like to make some comments below prior to publication.

Comments:

1) Lines 189-192: The authors describe “TauRD (KD = 0.54 μM) showed binding to Nup98-FG with an even higher affinity than full-length Tau (KD = 0.86 μM), whereas the binding of the N-term (KD = 0.1.25 μM) was low compared to full-length Tau (Figure 2D, E, F; Figure S2C, D)”, but is it correct? In Figure 2D, full-length tau appears to have the highest binding strength and TauRD the second highest. Also, these KD values do not appear to be the same as those shown in Fig. 2F. I am not an expert, and it is hard for me to understand. Please provide a more accurate explanation.

2) The question still remains as to whether the interaction with tau depends solely on GLEBS or on both FG and GLEBS. In this regard, perhaps the authors should have used Nup98-FG peptides without GLEBS instead of BSA, as well as other negatively and positively charged peptides as control ligands, to reach a more definitive conclusion. I would leave it to the authors to decide whether to perform such additional experiments for this manuscript.

3) Information about Nup98-FG and Nup98-FG/FS seems insufficient. Humans should have multiple splicing variants for Nup98. So, please specify which clone was used for the experiments or provide peptide sequences for Nup98-FG and -FG/FS as supplementary information.

4) Fig 3B is not cited in the text. Please cite it appropriately.

5) Line 367: Please explain about “K18”.

6) In the abstract, “PTMs” and “NCT” should be spelled out.

7) In Figure S5, the order of panels A, B, and C in the caption is different from the order in the drawings.

8) Also, in Figure S5, what do the gray circles with “%” represent?

9) This is not a comment, but a question. Does tau aggregation at the nuclear membrane really inhibit nucleocytoplasmic transport? Or does tau aggregation disrupt the permeability barrier function of the nuclear pore complexes?

Author Response

Point by Point answers to Reviewer 2

Diez et al. studied the interaction of Tau with Nup98 in the context of tauopathy, like Alzheimer’s disease. Using SPR and BLI, the authors measured association and dissociation of various Tau fragments with the N-terminal Nup98-FG or the full length of Nup98-SG mutant peptide and found TauRD and the phosphorylated Tau (either full length or delta K280) efficiently interact with Nup98-FG whereas acetylated Tau or oligomerized Tau did not. The data support the authors’ conclusions, and I am supportive of the publication of the paper. However, I would like to make some comments below prior to publication.

1) Lines 189-192: The authors describe “TauRD (KD = 0.54 μM) showed binding to Nup98-FG with an even higher affinity than full-length Tau (KD = 0.86 μM), whereas the binding of the N-term (KD = 0.1.25 μM) was low compared to full-length Tau (Figure 2D, E, F; Figure S2C, D)”, but is it correct? In Figure 2D, full-length tau appears to have the highest binding strength and TauRD the second highest. Also, these KD values do not appear to be the same as those shown in Fig. 2F. I am not an expert, and it is hard for me to understand. Please provide a more accurate explanation.

Authors: We thank the reviewer for addressing these points, which obviously caused confusion. The binding affinity for N-term is KD = 1.25µM (not 0.1.25 µM). We have corrected Table S2C, where the magnitude factor was wrongly written; see corrections in the Fig. S2 table C).

The KD values shown in Table S2C and Figure 2F demonstrate that TauRD (KD = 0.54 µM) and full-length Tau (2N4R; KD = 0.86 µM) are binding stronger than N-term tau (KD = 1.25 µM). This indicates that the C-terminus of Tau is substantially contributing to the interaction with Nup98-FG.

TauRD (KD = 0.54 µM) may bind stronger than full-length Tau (2N4R; KD = 0.86 µM), because the unstructured N-terminus (absent in TauRD) could screen certain interactions of the Tau repeat domain. In fact, Tau can adopt a ‘paper clip’ fold (Jeganathan, Biochemistry 2006), in which the unstructured N-terminal half folds back onto the repeat domain. This folding of the N- terminus can shield interactions of the repeat domain, which, for example, may explain the slower PHF aggregation of full-length Tau compared to TauRD constructs. We added a similar discussion to the text (Lanes 191-196).

In Figure 2D, we show that the BLI binding response is higher for 2N4R than for TauRD, indicating a stronger accumulation for the full-length 2N4R construct. The binding affinity (KD values), however, can only be extracted after fitting the raw data. The discrepancy between the values of Figure 2D and 2F arises from the inter-experimental variances that are dependent on how dense or sparse the tip is immobilized with Nup98-FG. Therefore, we compared only data that were achieved within one experiment (and BLI tip or SPR chip). This is stated in the methods parts.

2) The question still remains as to whether the interaction with tau depends solely on GLEBS or on both FG and GLEBS. In this regard, perhaps the authors should have used Nup98-FG peptides without GLEBS instead of BSA, as well as other negatively and positively charged peptides as control ligands, to reach a more definitive conclusion. I would leave it to the authors to decide whether to perform such additional experiments for this manuscript.

Authors: We thank the reviewer for this suggestion! It is indeed very interesting to further investigate the relevance of the GLEBS domain in more detail since this domain is important for RNA export from the nucleus. However, to produce such constructs and run the experiments would exceed the framework of our current study. We will certainly consider these experiments in the future though.

BSA is used in SPR experiments as an inert non-binding reference molecule in between Tau concentrations, which allowed us to determine the intrinsic thickness of the FG-Nup layer before applying the next Tau solution (Schoch et al. 2012; Schoch and Lim 2013; Kapinos et al. 2014).

3) Information about Nup98-FG and Nup98-FG/FS seems insufficient. Humans should have multiple splicing variants for Nup98. So, please specify which clone was used for the experiments or provide peptide sequences for Nup98-FG and -FG/FS as supplementary information.

Authors: We now provide the peptide sequences for Nup98-FG and Nup98-FG/FS as supplementary information (Figure S1D).

4) Fig 3B is not cited in the text. Please cite it appropriately.

Authors:  We added the citation to the text. Thank you for catching this mistake.

5) Line 367: Please explain about “K18”.

Authors: We apologize for the inconsistency by using “K18" instead of “TauRD”, which is our internal nomenclature for the TauRD construct and changed it accordingly in the manuscript.

6) In the abstract, “PTMs” and “NCT” should be spelled out.

Authors: This has been changed.

7) In Figure S5, the order of panels A, B, and C in the caption is different from the order in the drawings.

Authors: We thank the reviewer for pointing out this mistake. We changed the manuscript accordingly.

8) Also, in Figure S5, what do the gray circles with “%” represent?

Authors: We thank the reviewer for bringing up this point. In the bubble charts, the bubble size matches the %-age of the respective particle size in the preparation as determined by DLS. We added a better explanation to the figure legend of Figure S5.

9) This is not a comment, but a question. Does tau aggregation at the nuclear membrane really inhibit nucleocytoplasmic transport? Or does tau aggregation disrupt the permeability barrier function of the nuclear pore complexes?

Authors: This is a very good question, and according to our current knowledge, both scenarios are possible. We previously observed permeability barrier leakage in nuclei isolated from AD brain by Dextran exclusion assay (Eftekharzadeh et al., Neuron 2018). This correlated with Braak staging, meaning with the amount and percentage of tangle bearing neurons. In this case, aggregated Tau may lead to NPC leakage. However, we also observe NCT failure in cells without non-aggregated Tau at the nuclear envelope (Hochmair et al., EMBO J 2022), which may be accounted for transport impairment rather than leakage of the pore. In another study, we are currently investigating which factors of the NCT machinery can interact with Tau, and which precise part of the transport is impaired.

Reviewer 3 Report

The manuscript entitled ‘Phosphorylation but not oligomerization drives the accumulation of Tau with nucleoporin Nup98’ by Lisa Diez et al represents a very interesting manuscript where the authors investigated the molecular details of Tau:Nup98 interaction using the following methodologies, surface plasmon resonance (SPR) and bio-layer interferometry (BLI). The authors showed that this interaction is dependent of different Tau Forms. Therefore, a transient binding of (unmodified) Tau to Nup98-FG layers is mediated by the Tau repeat domain. Further, a slow association and deposition of phosphorylated Tau (P-Tau) at Nup98-FG is observed and theinteraction is blocked with pre-formed Tau oligomers.

In overall, I consider that the premise of this study is very interesting and important for the field, and I will perform some comments and suggestions.

Major concerns:

  1. The introduction should end with the objective of the present study.
  2. Regarding Figure 1D, have the authors confirmed the localization of p-Tau at NE using a NE marker, for instance Lamina A/C or any of the outer or inner membrane proteins?
  3. Regarding the following statements, ‘A consequence of NE distortions and Tau accumulation at the NE is the impairment of molecular transport into and out of the nucleus (lines 47-48’ what is the meaning of NE distortions??? Do you mean NE dysfunction? There is a recent paper about NE dysfunction in Aging.
  4. Could the authors explain the meaning of the following sentence: ‘This stress-related localization, considered as missorting [6], enables interactions with the nuclear envelope and NPCs (311-312).
  5. The molecular mechanism underlying P-Tau and FG-Nups association should be explored to clarify if P-Tau and FG-Nups association has a functional reason or is a consequence of Tau mislocation.

Author Response

Point by Point answers to Reviewer 3

The manuscript entitled ‘Phosphorylation but not oligomerization drives the accumulation of Tau with nucleoporin Nup98’ by Lisa Diez et al represents a very interesting manuscript where the authors investigated the molecular details of Tau:Nup98 interaction using the following methodologies, surface plasmon resonance (SPR) and bio-layer interferometry (BLI). The authors showed that this interaction is dependent of different Tau Forms. Therefore, a transient binding of (unmodified) Tau to Nup98-FG layers is mediated by the Tau repeat domain. Further, a slow association and deposition of phosphorylated Tau (P-Tau) at Nup98-FG is observed and the interaction is blocked with pre-formed Tau oligomers.

In overall, I consider that the premise of this study is very interesting and important for the field, and I will perform some comments and suggestions.

Major concerns:

The introduction should end with the objective of the present study.

Authors: we changed the introduction accordingly.

Regarding Figure 1D, have the authors confirmed the localization of p-Tau at NE using a NE marker, for instance Lamina A/C or any of the outer or inner membrane proteins?

Authors: Unfortunately, our effort to immunonstain AD brain sections for different Lamins (B1, B2, A/C) and NPC proteins was not successful at this time, and we will need to optimize our epitope unmasking and immunolabeling protocols for these targets. This would take multiple weeks. However, we recently performed and published co-labeling of Lamin B1, NPCs, and Tau in a human ‘Tau aggregation reporter’ cell line (HEK sensor cells) (Hochmair et al, EMBO. 2022). The respective reference is cited in the discussion (Lane 318).

Regarding the following statements, ‘A consequence of NE distortions and Tau accumulation at the NE is the impairment of molecular transport into and out of the nucleus (lines 47-48’ what is the meaning of NE distortions??? Do you mean NE dysfunction? There is a recent paper about NE dysfunction in Aging.

Authors: We thank the reviewer for pointing out this imprecision. We changed the wording to ‘NE deformations (lamin folds and invaginations)’ in the sentence before as this is what was observed in the cited reference. In the respective called-out sentence, we now use ‘NE dysfunction’. We hope this is now clearer described in the text.

Could the authors explain the meaning of the following sentence: ‘This stress-related localization, considered as missorting [6], enables interactions with the nuclear envelope and NPCs (311-312).

Authors: We changed the sentence in the manuscript to make better it understandable.

The molecular mechanism underlying P-Tau and FG-Nups association should be explored to clarify if P-Tau and FG-Nups association has a functional reason or is a consequence of Tau mislocation.

Authors: We agree that this is one of the key questions to be investigated next. We share the idea that Tau accumulation at the NE not necessarily has to be pathological but may in fact be a mechanism how neurons use Tau to block NCT in stress conditions, for example to inhibit transcription and translation. This idea is very exciting and controversial in the field, and we are currently working on investigating whether p-Tau association with the NE is functional or pathological, or both. The details of these experiments are confidential among people working on the project and collaborators, and we hope that the reviewer understands that we don’t want to disclose them. In any case, we are not able to provide data on the question of functional versus pathological Tau association with the NE in this current study, which focused on the molecular details and kinetics of Tau to Nup98 in NPCs. The herein presented data are very important for our understanding of this process though.    

Round 2

Reviewer 3 Report

Please find my comments about the revision of the manuscript entitled ‘‘Phosphorylation but not oligomerization drives the accumulation of Tau with nucleoporin Nup98’ by Lisa Diez et al The authors respond to all issues raised in my revision and performed the adequate alterations of the manuscript and the latter was significantly improved.

Overall, I believe that the manuscript is ready for publication.